# Stimulus-Evoked Activity Modulation of In Vitro Engineered Cortical and Hippocampal Networks

**DOI:** 10.3390/mi13081212

**Published:** 2022-07-29

**Authors:** Francesca Callegari, Martina Brofiga, Fabio Poggio, Paolo Massobrio

**Affiliations:** 1Department of Informatics, Bioengineering, Robotics and Systems Engineering (DIBRIS), University of Genova, 16145 Genova, Italy; francesca.callegari@edu.unige.it (F.C.); martina.brofiga@dibris.unige.it (M.B.); fabio.poggio@edu.unige.it (F.P.); 2ScreenNeuroPharm s.r.l., 18038 Sanremo, Italy; 3National Institute for Nuclear Physics (INFN), 16146 Genova, Italy

**Keywords:** cortical neurons, hippocampal neurons, electrical stimulation, Micro-Electrode Arrays, engineered neuronal networks, polydimethylsiloxane, microchannels

## Abstract

The delivery of electrical stimuli is crucial to shape the electrophysiological activity of neuronal populations and to appreciate the response of the different brain circuits involved. In the present work, we used dissociated cortical and hippocampal networks coupled to Micro-Electrode Arrays (MEAs) to investigate the features of their evoked response when a low-frequency (0.2 Hz) electrical stimulation protocol is delivered. In particular, cortical and hippocampal neurons were topologically organized to recreate interconnected sub-populations with a polydimethylsiloxane (PDMS) mask, which guaranteed the segregation of the cell bodies and the connections among the sub-regions through microchannels. We found that cortical assemblies were more reactive than hippocampal ones. Despite both configurations exhibiting a fast (<35 ms) response, this did not uniformly distribute over the MEA in the hippocampal networks. Moreover, the propagation of the stimuli-evoked activity within the networks showed a late (35–500 ms) response only in the cortical assemblies. The achieved results suggest the importance of the neuronal target when electrical stimulation experiments are performed. Not all neuronal types display the same response, and in light of transferring stimulation protocols to in vivo applications, it becomes fundamental to design realistic in vitro brain-on-a-chip devices to investigate the dynamical properties of complex neuronal circuits.

## 1. Introduction

The mammalian nervous system is characterized by the existence of an intrinsic spontaneous activity that displays complex patterns, such as oscillations, rhythms at well-defined frequencies, spikes, and bursts lasting from a tenth to hundreds of milliseconds [1,2,3,4]. The network organization is deemed to be the cause of the genesis of such dynamics, as demonstrated by several studies both at the in vivo and in vitro level ([5,6] and references therein). Furthermore, it is clear and well-established that the nervous system modulates its intrinsic spontaneous activity when it receives sensory stimuli [7,8]. Typically, the response to stimuli depends on the single-cell specialization and on the connectivity of the network itself: the stimulation of a hub neuron or of a highly-connected local cluster of neurons will produce different effects than the stimulation of sparse weakly connected assemblies. Therefore, it has emerged how two fundamental features can influence the response of a neuronal ensemble, namely, the neuronal type (e.g., cortical, hippocampal, thalamic, etc.) and the topological properties (e.g., degree of modularity or clusterization, existence of long-range connections and/or hubs, etc.) of the involved network.

A possible way to investigate the response of neuronal assemblies is to use dissociated neuronal cultures coupled to Micro-Electrode Arrays (MEAs). In this simplified experimental model, neuronal networks can freely form synaptic connections during their natural development by self-organizing in complex topologies such as rich-club [9] and scale-free [10]. However, its great potential is that it is possible to engineer the substrate where neurons grow to design well-defined circuits by promoting topological properties such as modularity [11], ordered lattices [12], or pushing the directionality of the connections [13]. This engineering process guarantees the modulation of the emerging neuronal dynamics [14] as well as precise control over the interacting neuronal types [15]. Since the early 2000s, these models were used to characterize the neuronal response to different electrical stimulation protocols, which differ in the frequency and (partially) in the amplitude of the delivered stimuli.

High-frequency stimulation protocols were tested on dissociated neuronal networks coupled to MEAs, with the aim to induce forms of long-term plasticity (potentiation and depression). Typically, plasticity protocols exploit tetanic stimulations. From the pioneering work of Jimbo and coworkers, who proved that it was possible to induce long-lasting changes in the network responses [16], variations in these high-frequency stimulation patterns were used for learning protocols [17] or with the coexistence of a low-frequency stimulation pulse to induce long-term network potentiation [18]. This last protocol was also able to induce a reshaping of the underlying functional connectivity by increasing the number of functional connections [19].

At the same time, stimulation protocols based on the delivery of low-frequency pulses also was investigated to appreciate the modes of response of a neuronal network. By providing a low-frequency stimulation (less than 1 Hz), Eytan and coworkers found a transient enhanced response of the network as a function of the stimulated subnetwork (i.e., stimulation sites) [20]. Some years later, Vajda and colleagues demonstrated that low-frequency stimulation shaped the patterns of spontaneous electrophysiological activity by changing features such as the shape of the network bursts [21]. More recently, the interplay between low-frequency stimulation patterns and the underlying functional connectivity was investigated, too [22].

In the present work, we investigated the low-frequency response of two different neuronal populations (namely cortical and hippocampal ones) plated over MEAs with a modular connectivity. The development of three interconnected assemblies was obtained thanks to a polymeric mask (realized in polydimethylsiloxane, PDMS) reversibly coupled to the surface of the MEAs. We investigated and compared the stimuli-evoked response of the cortical and hippocampal populations as a function of the location of the stimulation sites (i.e., stimulated assembly). This analysis was carried out by taking into account the different synaptic excitatory responses of the network, namely, the early phase—mediated by the α-amino-3-hydroxy-5-methyl-4-isoxazolepropionic acid (AMPA) receptors, lasting no more than 35 ms; and the late phase—N-methyl-D-aspartate (NMDA) mediated, lasting hundreds of milliseconds.

Our results show that the cortical assemblies are generally more reactive than the hippocampal ones to the delivery of low-frequency stimulation protocols. Moreover, although both the cortical and hippocampal cultures exhibited a fast (<35 ms) response, this did not propagate uniformly in the hippocampal networks. In addition, we found that only cortical assemblies displayed a significant late (35–500 ms) response. The achieved results suggest the need to develop brain-on-a-chip models as test beds for the characterization of neuronal dynamics and for studying the efficacy of stimulation protocols to interact with the neuronal structures. Using the same principles and observations, devices such as the one we employed in the present work could help getting more reliable results, also in chemical-based modulations (neuromodulation). To put it in perspective, to achieve a more comprehensive knowledge of the nervous system, next-generation devices should include multi-functional components for the monitoring not only of the electrophysiological activity but also of the relevant parameters relative to the delivery of the selected chemicals, and for the fine control over the experimental environment by means of perfusion motorized systems [23].

## 2. Materials and Methods

### 2.1. Polymeric Device

A three-compartment polydimethylsiloxane (PDMS) mask was used for the experiments. It consists of one larger chamber (6.5 mm wide, 4.3 mm long, with 16 μL capacity) and two smaller ones (3.4 mm in diameter and with 8 μL capacity) interconnected with arrays of 28 regularly spaced (50 μm) microchannels (Figure 1A) [24]. The microchannel dimensions (10 μm wide, 250 μm long, and 5 μm high) prevent the migration of cell bodies among the compartments while allowing the crossing of neurites only [25].

The mask was fabricated by conventional soft lithography molding techniques (Scriba Nanotechnologies S.r.l., Bologna, Italy). The SU-8/silicon master was obtained by consecutive steps of UV photolithography performed on different photoresistant layers of different thicknesses. The pattern of microchannels was created upon a 5-μm-thick SU 2005 layer. The cell chambers structures were aligned and superimposed on a 300-μm-thick SU 2075 layer. The prebaking and postbaking time were optimized to preserve the structure of the microchannels. After development, a thin layer (400 nm) of polyacrylic acid was spin-coated upon the master to enhance the peel-off of the PDMS replica.

The replicas were obtained by pouring a PDMS solution (curing agent 1:10 *w*/*w* in prepolymer) on the mold followed by heat curing in dry oven at 80 °C for 20 min. The masks were then aligned and reversibly bounded to planar MEAs (Figure 1B) with 120 electrodes (30 μm in diameter and 200 μm in spacing, arranged in a 12 × 12 array). In order to obtain a reversible sealing between the PDMS mask and the MEA substrate, a plasma oxygen treatment (60 W for 10 s) was applied to the surfaces to be coupled. Then, the treated surfaces were coupled and the achieved device (i.e., MEA with PDMS mask) was placed at 60 °C for 1 hour. To ensure a good hydrophilization of the microchannels, after sterilization the devices underwent oxygen-plasma treatment (120 W for 60 s).

### 2.2. Cell Culture Preparation

Primary cultures of Sprague-Dawley rat embryonic (E18-E19) cortical and hippocampal neurons were obtained by enzymatic digestion of hippocampi and cortices with a Trypsin (0.125%) and DNAse (0.03%) solution (in Hanks’ solution). The process was quenched by adding fetal bovine serum (FBS, 10%) supplemented medium (Neurobasal). Single cells were obtained with a mechanical trituration of the tissues with a fine-tipped Pasteur pipette. The cell suspension was diluted in serum-free medium supplemented with 2% B-27 Supplement, 1% stable l-Glutamine, and 1% PenStrep. Glia proliferation was not prevented as glia cells are known to be necessary for the development of healthy neuronal populations [26,27]. Cells were plated on 120-channel MEAs, which were previously coupled to masks (Section 2.1), sterilized in dry oven for 3 h, plasma treated, and coated with poly-l-ornithine (100 μg/mL).

Cells of the same neuronal type were plated on MEAs at a defined density (1500 cell/mm^2^ for cortical and 1300 cell/mm^2^ for hippocampal cells) to avoid effects on the network development due to different plating densities [2]. The samples were incubated at 37 °C, with 5% CO_2_ and 95% humidity. After 5 days, and twice a week afterward, half volume of the medium was replaced with BrainPhys medium, supplemented with 2% NeuroCult SM1, 1% Glutamax, and 1% PenStrep solution. Neurons were let to organize into a morphologically and functionally mature network for 3 weeks before experiments were performed (Figure 1B, *right*). To maintain the same experimental conditions during the recording phase, we controlled both the temperature (37 °C) and the flow of humidified gas (5% CO_2_, 20% O_2_, 75% N_2_) supplied to the cell culture environment.

### 2.3. Ethical Approval

The experimental protocol for in vitro cultures was approved by the European Animal Care Legislation (2010/63/EU) and the Italian Ministry of Health, in accordance with D.L. 116/1992 and the guidelines of the University of Genova (Prot. 75F11.N.6JI, 08/08/18).

### 2.4. Experimental Protocol

The electrophysiological activity of the neuronal networks was recorded at DIV 18–19 using the MEA2100 system (Multi Channel Systems, MCS, Reutlingen, Germany) with a sampling frequency of 10 kHz. Recordings were performed outside the incubator. To minimize the thermal and mechanical stress on the cells, excessive evaporation, and unphysiological changes to the pH of the cultures, the MEA lodging was heated (37 °C) and a constant slow flow of humidified gas (5% CO_2_, 20% O_2_, 75% N_2_) was supplied to the cell culture environment. MC_Rack software (MCS) was used for data recording and for delivering the electrical stimulation on individual electrodes.

Each experimental session lasted 25 min. First, 10 min of spontaneous activity was recorded to evaluate the basal dynamics (i.e., presence of spikes and bursts, number of active electrodes, etc.) of the network. Then, one electrode for each compartment was chosen to be stimulated for 5 min. The selection of the stimulated electrodes was based on their initial spontaneous activity: only the more active electrodes were chosen as possible stimulating sites, indicating that they were functionally connected to the rest of the network. The stimulating pulse was biphasic (positive phase first); it lasted for 400 µs with a 50% duty cycle and was sent at a frequency of 0.2 Hz. The choice of such a low-frequency stimulation was driven by conventional electrophysiological stimulation protocols based on test stimuli that induce non-plastic responses [16,18]. Overall, the amplitude of each pulse was 1.5 V peak-to-peak relative to the reference electrode.

To make sure that the order of the stimulation did not influence the results, the first compartment to be stimulated was changed for each experimental session.

### 2.5. Dataset

The dataset used in this work consists of spontaneous and stimulus-evoked recordings from 8 different cultures (4 cortical (Cx) and 4 hippocampal (Hp) networks) coming from two preparations. All cultures were recorded at 18–19 day in vitro (DIV).

### 2.6. Data Analysis

Electrophysiological activity was characterized offline by using customized analysis tools developed in MATLAB 2021a (The Mathworks, Natick, MA, USA). The first step was the identification of the spikes exploiting the Precise Timing Spike Detection (PTSD) method [28]. We set three parameters: (i) the peak lifetime period to 2 ms; (ii) the refractory period to 1 ms; and (iii) the adaptive threshold for each electrode to 8 times the standard deviation of thermal and biological noise. Data were not spike sorted, since during a bursting event, a global increase of the activity produces a fast sequence of spikes with different and overlapping shapes, which makes the sorting difficult and unreliable [2].

To evaluate the spontaneous activity of the network, we analyzed the basal spiking activity in terms of Mean Firing Rate (MFR). An electrode was discarded from the analysis if the MFR was lower than 0.1 spike/s. In addition, we characterized the bursting activity. For this purpose, we identified bursts by applying the string method devised in [29], by setting the minimum number of spikes in a burst to 5, and the maximum inter spike interval into bursts at 100 ms. Then, we extracted the following parameters: the Mean Bursting Rate (MBR), i.e., the mean number of bursts per minute; and the Burst Duration (BD), i.e., the temporal duration of the bursts.

To obtain quantitative information about the stimulus-evoked activity, we computed the Post-Stimulus Time Histogram (PSTH), which represents the impulsive response of each electrode to the electrical stimulation. The PSTH was computed considering time windows of 600 ms that follow each stimulus. Then, we divided that interval into 4 ms bins and counted the number of spikes that occurred in each time interval. Eventually, we normalized the obtained histogram by dividing the number of spikes detected for each bin by the number of stimuli. If an electrode had a PSTH with area lower than “1”, that channel was removed from the statistics because it was considered inactive (i.e., unable to evoke minimal response to stimulation). Furthermore, in order to use the PSTH area as a measure of the response of each electrode and to be able to compare the data among the different networks and configurations, we normalized the PSTH areas over the maximum obtained value.

The response evoked by an electrical stimulation is typically characterized by two components: an *early* and rapid one, which occurs in a time range between 0 and 35 ms after the stimulus; and a *late* and slower one, which occurs around the 35–500 ms. Previous works demonstrated their presence due to the AMPA- and NMDA-mediated transmission by performing selective blocking with antagonist molecules such as amino-5-phosphonovaleric acid (APV) and cyanquixaline (CNQX), respectively [30]. In order to identify and characterize these responses of the different neuronal populations, we developed a new method to distinguish the *early* component from the *late* one. We classified the PSTH for each electrode with the *k*-mean algorithm and the silhouette analysis (to identify the optimal number of classes). After the PSTH classification, we computed the average response for each class *j* (obtained by averaging the responses of each active electrode belonging to the same class), and we smoothed it by applying a moving average (*PSTH_smooth,_* temporal bin of 50 ms). Then, we detected the local maxima of *PSTH_smooth_*: if a peak was detected within the first 52 ms (*early*, *x*_1_), a second one with at least 10% of the prominence of the first peak was sought between 52 and 600 ms (*late*, *x*_2_). We searched for the local minimum (*x_min_*) within the range of the two peaks, and we evaluated the separation between them following the approach devised in [31]:(1)$$s_{j}=\frac{PSTH_{smooth,j}\left(x_{min}\;\right)}{\sqrt{PSTH_{smooth,j}\left(x_{1}\right)PSTH_{smooth,j}\left(x_{2}\right)}\;}$$

If *s_j_* exceeded an empirical threshold set at 0.3, *x_min_* was considered the optimal time threshold for dividing the *early* and *late* responses for the *j*-class. Otherwise, the class was characterized only by the *early* response, and its latency value for the *late* response was set to a value higher than the observation window (i.e., 650 ms).

### 2.7. Statistical Analysis

Data are expressed as the mean ± standard deviation of the mean. Statistical analysis was performed using a nonparametric Kruskal–Wallis test since data did not follow a normal distribution (evaluated by the Kolmogorov–Smirnov normality test). Differences were considered statistically significant when *p* < 0.05. The box plots representation indicates the 25–75 percentile (box), the standard deviation (whiskers), the mean (square), and the median (line) values.

## 3. Results

The presented results aim to demonstrate that different cell types in vitro respond differently when they receive external stimulation. In the following sections, we explored the intrinsic activity patterns of the two considered cell types for the present study, namely, cortical (Cx) and hippocampal (Hp) cells (Section 3.1). Then, we analyzed the response produced by the delivery of a low-frequency stimuli on the two considered configurations and evaluated their differences both in terms of “amplitude” and of propagation of the evoked activity (Section 3.2).

### 3.1. Interconnected Cortical and Hippocampal Assemblies Display Spontaneous Activity

Firstly, we evaluated the spontaneous electrophysiological activity of the networks to characterize the intrinsic parameters of each network and their differences. Qualitatively, both cortical and hippocampal (Figure 2A, shades of red and green, respectively; the shades of color used in the raster plot representation identify their position in the MEA configuration) dynamics displayed the coexistence of a random spiking activity and highly packed bursts. This ensures that the neuronal populations under consideration had established an adequate, functionally connected network able to maintain sustained spiking and bursting patterns of activity. Without this condition, any stimulation protocol would be ineffective. As the site of simulation was not significant for these analyses, the activity of all three compartments was condensed and considered as one.

Generally, the cortical (light grey) populations displayed higher spiking activity (Figure 2B, *p* = 0.03) and scattered data points with respect to the one of hippocampal (dark grey) networks, which on the other hand had very little variability.

Moving on to the characterization of the bursting dynamics of the two populations, we observed that the increase in the firing did not induce a difference in the bursting rate (Figure 2C). However, the different cell types of the two networks brought a change in the shape of the bursts, as they were significantly longer in the Cx case (Figure 2D, *p* = 0.03).

### 3.2. Low-Frequency Electrical Stimulation Produces Different Response Patterns in Cortical and Hippocampal Populations

After the initial assessment of the spontaneous activity of both populations, the stimulus-evoked activity was evaluated by means of the PSTH (cf. Section 2.6). Figure 3 shows normalized PTSHs for a representative cortical (top row) and hippocampal (bottom row) network stimulated in three different electrodes, each belonging to a different compartment (the site of simulation is indicated as a lighting near the related compartment). In particular, the cortical interconnected sub-populations exhibited a clear and marked early response followed by a long late response, independently from the choice of the location of the stimulated channel, as demonstrated by the double peak in the PSTH (Figure 3A–C). On the contrary, the hippocampal sub-networks apparently presented only an early response, which in most cases decayed within the first 200 ms (Figure 3D–F). Moreover, the considered experiments showed that the Cx cultures seemed to be more responsive to external electrical stimulation than the Hp configuration in terms of amplitude of the PSTH areas.

To prove this observation, we computed the area under the PSTH for the two populations. Figure 4A,B depict the color- and height-coded area computed for each electrode in two representative experiments, stimulated in the electrode depicted with a red dot. Qualitatively, it was evident from this example that the Hp population had a lower response in terms of amplitude (Figure 4B) than the one observed in the Cx counterpart (Figure 4A). From the plots, we observed that the representative Hp network displayed a peak of the PSTH area of 40 spikes, while in the cortical population we reached a peak of 120 spikes. Moreover, only few electrodes adjacent to the stimulated site had a high response in the Hp assembly, whereas the cortical response was rather uniform throughout the network. This consideration was confirmed by the cumulative normalized PSTH areas for Cx and Hp (Figure 4C), which were computed by considering the response of the two populations in any configuration. It highlighted a significant difference (*p* = 9.9 × 10^−9^) between the two populations.

Next, we investigated whether the site of stimulation had any effect on how much the network responded. Generally, the position of the stimulus did not produce significant variations in the amplitude of response of the three compartments. However, it had some effect when comparing the two studied configurations. In fact, when the stimuli were delivered in the small compartment, the Cx and Hp responses were statistically different, either only in the non-stimulated (Figure 4D,E) or in all (Figure 4F) the compartments. The larger stimulated compartment instead produced a difference between Cx and Hp only in one of the two smaller compartments.

Another feature that characterizes the response mode of the neurons is how the signal propagates throughout the population. The modular topology of the networks induced by the presence of the PDMS mask and the use of two neuronal types could produce changes in the transmission of the evoked activity, consequently modifying the velocity of involvement of the electrodes. To quantify these possible differences, we extracted from the PSTH the *early* and *late* latency components, as described in Section 2.6. The first interesting result was that most of the Hp networks did not exhibit a late response. On the other hand, the Cx cultures showed both a fast uniformly distributed response (Figure 5A, *left*) and a slow response, whose latency values increased with the distance from the stimulation electrode (Figure 5A, *right*). It is interesting to notice that the uniformity of propagation was not as evident in the Hp configuration (Figure 5B), where the direction of propagation of the signal was not clearly recognizable.

This observation was confirmed by comparing the cumulative latency values for the two populations (Figure 5C,D). In particular, for each stimulation, both the early and the late latency mean values were computed for each compartment. We observed that the early response (Figure 5C) was evoked in every compartment for each stimulus (number of evoked responses: *n_Cx_* = *n_Hp_* = 36) in both the Cx and the Hp populations, which however manifested some differences. In fact, the Hp networks were significantly faster in this phase (*p* = 8.7 × 10^−5^). Regarding the late component (Figure 5D), it was present in almost all cases in the Cx (*n_Cx_* = 30), while it was almost absent in the Hp networks (*n_Hp_* = 6). Consequently, a statistical difference in the latency values between the two populations was found (*p* = 2.3 × 10^−9^). Considering the previous results regarding the stimuli position in the MEA layout, we again analyzed the response time, depending on the position of the stimulated electrode (Figure 5E,F). We did not find statistical differences within the same population, although the stimulated compartment generally had slightly lower latency values. However, we observed significative differences between the Cx and Hp populations, especially in the late response. If in the early latency values the cortex resulted to be statistically slower only in three cases (Figure 5F), the comparison of late latency values gave rise to significant differences in all the cases.

## 4. Discussion

In vivo, small/medium-sized neuronal networks are continuously bombarded by inputs coming from different sources, ranging from external stimuli (e.g., chemical, electromagnetic, nociceptive, cognitive, etc.) to synaptic inputs projected by other neuronal assemblies. This huge number of stimuli continuously triggers and shapes the spontaneous electrophysiological activity of the nervous system. The processing of external inputs depends not only on the features of the presented patterns of stimulation but also on several internal neuronal and network variables defining the dynamical state of the circuit [32]. Although many studies (both theoretical and experimental) have focused on understanding the input–output functions that regulate the dependency between the delivered stimulation and evoked response [33,34,35,36,37], the whole picture is not completely clear yet.

In the present work, we exploited the controllability of an engineered in vitro system made up of interconnected neuronal networks coupled to Micro-Electrode Arrays (MEAs) to explore the role of two of the aforementioned internal neuronal variables on connectivity and neuronal heterogeneity. We applied low-frequency stimulation protocols to cortical and hippocampal neurons, topologically organized by means of a PDMS mask reversibly bounded to the active area of the MEA, to guarantee the segregation of the cell bodies. The role of these physical constraints (with or without microfluidics properties) for the modulation of the spontaneous activity of dissociated neuronal networks has been extensively studied [11,38,39,40]. In the present work, a mask inspired by the work of [24] was employed to define a neuronal network made up of three interconnected subpopulations to quantify the effect of a low-frequency electrical stimulation. The strategy to use polymeric masks to segregate neuronal populations and modulate their connectivity is a well-consolidated technique that allows imposing a directionality to connectivity (e.g., [13,41]) or to interface different neuronal types (e.g., [42,43]). In the present work, our polymeric mask had only the role to confine three sub-populations without forcing the directionality of their connections. The three-compartment mask devised in Figure 1, was used for both cortical and hippocampal cultures. At this stage of maturation (18–19 DIV), the network is already well-structured and a dense connectivity among the compartments is already present [24], causing a sustained spontaneous spiking and bursting activity (Figure 2). We found that cortical assemblies are more reactive than hippocampal ones to the delivery of low-frequency stimulation protocols (Figure 4C). We can speculate that such behavior could be explained by two factors: an intrinsic higher membrane excitability of cortical neurons or a different network organization, which facilitates synaptic integration in the case of cortical ensembles. In addition, we found that both cortical and hippocampal neurons exhibited a fast (<35 ms) (AMPA mediated) response that, however, did not uniformly propagate over the MEA in the hippocampal networks (Figure 5). Nonetheless, only cortical assemblies displayed a significant late (35–500 ms) response, typically mediated by the presence of excitatory NMDA synaptic receptors (Figure 5). In conclusion, the achieved results suggest the relevance of the neuronal target when an electrical stimulation experiment is designed and performed: the fundamental evidence emerging from the present work is that not all neuronal types display the same modes of response. This becomes fundamental when electrical stimulation protocols are transferred to in vivo experiments. Nowadays, the beneficial effects of electrical stimulation to cure or to reduce the symptoms of some brain impairments are well recognized: injuries [43], strokes [44], and tremors [45] are only some examples that have been treated by means of ad hoc electrical stimulation protocols in the last years. However, if the effect of an external stimulation on the electrophysiological activity and on the cognitive/motor functions is evident, the reasons beneath are still not completely understood [46]. Realistic in vitro models based on the concept of brain-on-a-chip could prove to be a valid support to solve this important question [47]. These hybrid devices (of which the system used in the present work is a simple example) allow performing experiments in a controlled environment, whose complexity should take into account the relevant intrinsic variables, such as the presence of different interacting neuronal populations [15,42], or the existence of topological features in the three-dimensional space [48].

## Figures and Tables

**Figure 1 micromachines-13-01212-f001:**
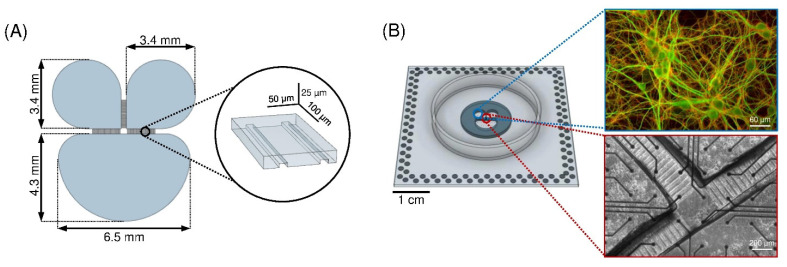
(**A**) Sketch of the silicon master (negative mold of the PDMS mask) used to create the polymeric device. In the inset, the dimensions of the microchannels of the PDMS mask are reported. (**B**) PDMS mask reversibly bonded to the MEA surface. (***Right*, *top***) Close-up of the cortical network (DIV 18) inside the compartment where microtubule-associated protein 2 (MAP2, green) and the vesicular glutamate transporter (Vglut1, red) are labeled. (***Right*, *bottom***) Close-up of the channel region of a cortical network organized in three interconnected sub-populations at DIV 18.

**Figure 2 micromachines-13-01212-f002:**
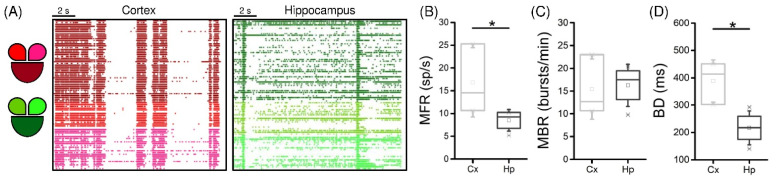
Spontaneous network activity of interconnected cortical (Cx) and hippocampal (Hp) assemblies: (**A**) 16 s of spontaneous activity of cortical (shades of **red**) and hippocampal (shades of **green**) interconnected populations; (**B**) MFR; (**C**) MBR; (**D**) BD of the cortical (**light grey**) and the hippocampal (**dark grey**) populations (* refers to 0.01 < *p* < 0.05; Kruskal–Wallis nonparametric test).

**Figure 3 micromachines-13-01212-f003:**
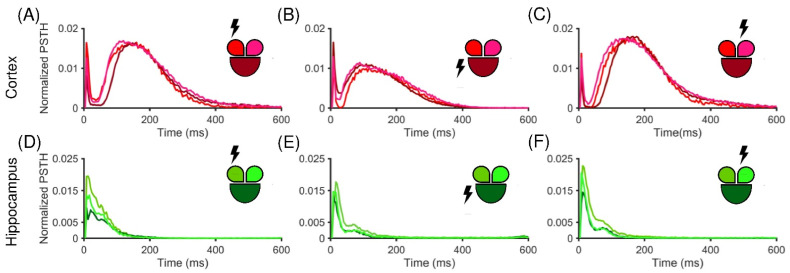
Normalized PSTH traces of a representative cortical (**A**–**C**) and hippocampal (**D**–**F**) network. The insets show the color-coded configurations. The stimulated compartment is indicated with a lightning sign in the inset. PSTH generated for (**A**) the small left, (**B**) the big central, and (**C**) the small right stimulated compartment in the cortical configuration. (**D**–**F**) The hippocampal counterpart in the same conditions.

**Figure 4 micromachines-13-01212-f004:**
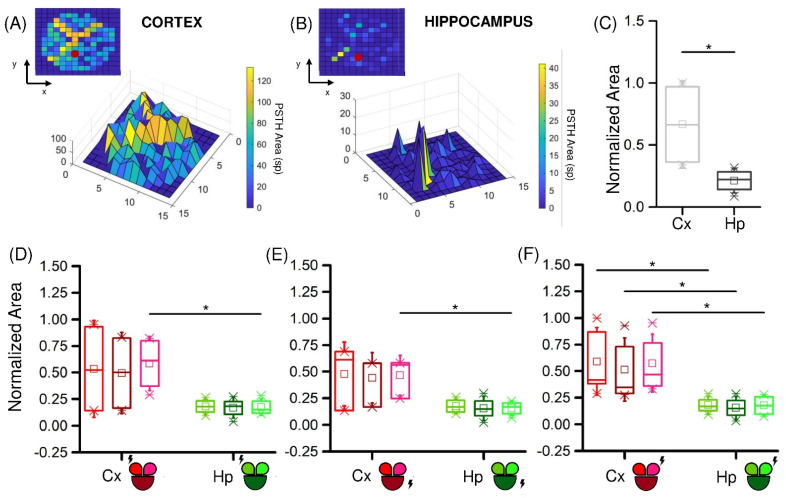
Cortical and hippocampal assemblies display different responsiveness when stimulated by low-frequency stimuli. Color- and height-coded representation of the PSTH area of a representative experiment with (**A**) Cx and (**B**) Hp neurons. (**C**) PSTH normalized area distributions of the Cx (**light grey**) and Hp (**dark grey**) assemblies. (**D**–**F**) Relevance of the site of stimulation (highlighted with a lightning sign) in cortical (**red**) and hippocampal (**green**) networks. (* refers to 0.01 < *p* < 0.05; Kruskal–Wallis nonparametric test).

**Figure 5 micromachines-13-01212-f005:**
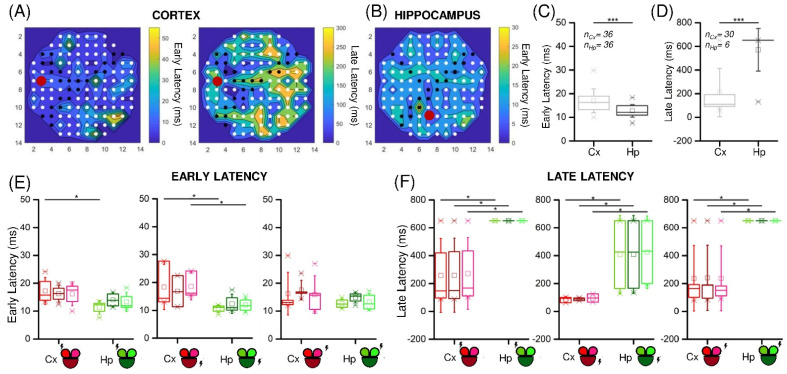
Cortical and hippocampal assemblies display different latency values when stimulated by low-frequency stimuli. Color-coded representation of the early and late (if present) latency values of representative (**A**) Cx and (**B**) Hp experiments. The red dot indicates the stimulated electrode. (**C**,**D**) Early and late latency cumulative values of Cx (**light grey**) and Hp (**dark grey**) assemblies. *n_Cx_* and *n_Hp_* indicate the number of evoked responses in the Cx and Hp configurations, respectively. (**E**,**F**) Relevance of the site of stimulation (highlighted with a lightning sign) in cortical (**red**) and hippocampal (**green**) networks in early and late latency values, respectively. (* refers to 0.01 < *p* < 0.05, *** to *p* < 0.001; Kruskal–Wallis nonparametric test).

## Data Availability

The peak trains of the entire dataset of this paper as well as the customized MATLAB functions used to analyze the data have been deposited in Zenodo. The DOI of the deposited data and code reported in this paper is: https://doi.org/10.5281/zenodo.6882374.

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
