# Peer review of "Stimulus-Evoked Activity Modulation of In Vitro Engineered Cortical and Hippocampal Networks"

_micromachines, 2022, doi:10.3390/mi13081212_

Round 1

Reviewer 1 Report

This is an interesting article that explores the impact of electrical stimulation on compartment-engineered neuronal cultures grown on MEAs. The article is clear and well-written and deservers publication in Micromachines, but some issues need to be addressed before publication.

1.The introduction is clear and well-oriented, but the description of the neuroengineering efforts on lines 46-48 is imprecise. Some referred works use indeed physical constraints to tune network topology, while others, especially Ref. [11], explain that non-trivial topological traits such as rich-club emerge from self-organization. The differences between directed topologies and self-organized ones such be clarified. Additionally, some of the engineering efforts have been conducted in calcium imaging experiments, and could be included there since authors really made an effort to relate physical structure with functional traits (e.g. Yamamoto et al., Impact of modular organization on dynamical richness in cortical networks, Science Advances 2018).

2. Additionally, related to the introduction, the concept of growing neurons in compartments and connecting them through small tunnels is not really novel. Particularly in MEA-based experiments, the authors should credit similar efforts conducted in the past, and that actually thoroughly analyzed functional aspects, such as ChihHsiang Chang et al., “Coupling of in vitro Neocortical-Hippocampal Coculture Bursts Induces Different Spike Rhythms in Individual Networks”, Front Neurosci 2022; Thomas B. DeMarse et al., “Feed-Forward Propagation of Temporal and Rate Information between Cortical Populations during Coherent Activation in Engineered In Vitro Networks”, Front Neural Circ 2016; L. Pan et al., “An In Vitro Method to Manipulate the Direction and Functional Strength Between Neural Populations”2, Front. Neur Circ.  2015; and others. The authors should discuss the novelty/differences of the present work as compared to those studies.

3. I don’t really understand the description of the PDMS mold. It seems that, on Figure 1A, the light-gray areas will be hollow in the final design, the white areas will be PDMS, and the dark-gray ones will be PDMS with channels for neurites development. Is that correct? A bit more of explanation in the caption would be helpful. Also, the inset of Figure 1A is too small. One cannot read the depicted dimensions. Authors could make the figure as 2x2 panels, with figure 1a and inset covering the top row.

4. The concept of “reversibly bonded” is not clear.

5. Figure 1C is good to see the channels, but one cannot see the neurons there.

6. The authors describe the presence of spiking activity and bursting. However, in figure 2, one can also see the presence of network bursts, i.e. the coordinated activation of most of the neurons in the network. Maybe the authors could include additional analyses to compare these network bursts between Cx and Hp cultures, or upon stimulation.

7. The PSTH concept to evaluate the response of the network upon stimulation seems a good approach, but I wonder how the authors know that the slow response (second peak in figure 3) is caused by the stimulation and it’s not an artifact caused by the ever-present spontaneous activity. For instance, the authors could use CNQX to reduce the level of spontaneous activity and apply similar stimulations to see if the peaks are maintained. Also, I think the figure 3 needs to be accompanied of raster plots, so the reader can judge how the networks react to stimulation.

7. The Cx and Hp networks seem ad hoc modular, but the three compartments in either case seem to activate in a coherent manner (judging by figure 3 and figure 5), which I think is due to a strong interconnectivity between compartments. Have the authors thought about reducing the interconnectivity to shape a richer repertoire of activity patterns (e.g., as in Yamamoto et al., Science Advances 2018).

8. I found very distracting the drawing of the electrodes in Figure 5A,B. Maybe they can be placed with some transparency, or be removed (except the stimulating one) so one can better appreciate the activation pattern.

9. The authors claim at the beginning of the discussion (line 356) that they explored connectivity, but different connectivity schemes between the 3 compartments was not explored. Also, I think the authors should comment on the degree of reproducibility of the results. Were different cultures carried out and the results compared? In general, I had the feeling that all the potential of the data was not used, such as functional connectivity analysis, as other studies did.

10. Modularity is mentioned in line 360, but the authors do not quantify it. I assume that it is beyond the scope of their work to carry out a functional connectivity analysis and infer functional modularity, but I think they could elaborate a bit more on the fact that they can do it and refer to studies where such an analysis has been done, such as the references mentioned in point 2.

Reviewer 2 Report

This study is interesting, but the authors need to address the following questions before the paper is ready for publication.

1. Please provide specific methods for culturing embryonic cortical and hippocampal neurons.

2. How to ensure that the state and density of cortical and hippocampal neurons are similar when stimulated?

3. How do the authors evaluate and distinguish between MBR and MFR?

4. On what basis did the author choose 0.2Hz low-frequency electrical stimulation? Do cortical and hippocampal neurons show different firing characteristics at other stimulation frequencies?

Round 2

Reviewer 2 Report

The authors have addressed all my questions. I don't have any further questions.